# Pivotal Role of the Granularity Uniformity of the WO_3_ Film Electrode upon the Cyclic Stability during Cation Insertion/Extraction

**DOI:** 10.3390/nano13060973

**Published:** 2023-03-08

**Authors:** Zhaocheng Zhang, Haoyuan Chen, Zicong Lin, Xiongcong Guan, Jiong Zhang, Xiufeng Tang, Yunfeng Zhan, Jianyi Luo

**Affiliations:** 1School of Applied Physics and Materials, Wuyi University, Jiangmen 529020, China; 2School of Civil Engineering and Architecture, Wuyi University, Jiangmen 529020, China; 3Research Center of Flexible Sensing Materials and Device Application Technology, Wuyi University, Jiangmen 529020, China

**Keywords:** film electrode, granularity uniformity, cyclic stability, cations inserting/extracting, stress distribution

## Abstract

Delicate design and precise manipulation of electrode morphology has always been crucial in electrochemistry. Generally, porous morphology has been preferred due to the fast kinetic transport characteristics of cations. Nevertheless, more refined design details such as the granularity uniformity that usually goes along with the porosity regulation of film electrodes should be taken into consideration, especially in long-term cation insertion and extraction. Here, inorganic electrochromism as a special member of the electrochemical family and WO_3_ films as the most mature electrochromic electrode material were chosen as the research background. Two kinds of WO_3_ films were prepared by magnetron sputtering, one with a relatively loose morphology accompanied by nonuniform granularity and one with a compact morphology along with uniform particle size distribution, respectively. Electrochemical performances and cyclic stability of the two film electrodes were then traced and systematically compared. In the beginning, except for faster kinetic transport characters of the 50 W-deposited WO_3_ film, the two electrodes showed equivalent optical and electrochemical performances. However, after 5000 CV cycles, the 50 W-deposited WO_3_ film electrode cracked seriously. Strong stress distribution centered among boundaries of the nonuniform particle clusters together with the weak bonding among particles induced the mechanical damage. This discovery provides a more solid background for further delicate film electrode design.

## 1. Introduction

Discovery of cations transporting and accumulating in solid electrodes (i.e., establishment of the intercalation theory [1]) has greatly propelled electrochemical technology, making spurts of progress. High-performance ion batteries [2], metal batteries [3], supercapacitors [4], and many other electrochemical devices [5] are constantly emerging. Inorganic electrochromism, as a special member of the electrochemical family, combines sustainable, reversible color changes for certain electroactive materials with cation insertion and extraction process upon the application of an electrical voltage [6]. Being capable of dynamically modulating heat and light on demand, electrochromic (EC) technology is highly expected as a promising solution for energy conservation and life-quality improvement, and it has captured much attention for decades [7,8,9]. With broader and broader applications in light adjustment [10], displays [11], electronic devices, and many others [12,13,14], two developing trends have formed in EC technology. On one hand, in academia, all-solid-state, more colorful, more functional, and faster-responding high-performance EC materials and technologies have been consistently advancing [15,16,17]. On the other hand, in the industrializing process, better stability, lower costs, and composite electrolyte and monolithic device optimization were explored [18,19,20].

WO_3_ EC films are most mature in application, originating from large optical modulation, low cost, great cyclic stability, as well as facile synthesis [8], and numerous efforts have been made to further improve their performances through doping [21], compounding with Nb_2_O_5_ or V_2_O_5_ [22,23], and other ways [24]. Undoubtedly, cyclic stability is the most decisive factor in real use. For WO_3_ films, the cyclic failure is generally attributed to the following two aspects: (1) electrochemical failure due to irreversible injection and massive accumulation of Li ions in the film [25]; and (2) volume expansion during cation injection/extraction, resulting in mechanical cracking of the film [26]. Either way, improving cyclic stability requires delicate design of the film electrode, where morphology of the WO_3_ film is the most important both for the electrochemical performances and the cyclic stability, which has been continuously optimized through modulating the heat-treatment temperature [27], the annealing method [28], preparation parameters [29,30], the crystallization, and nanostructures [31,32]. As a matter of fact, in the whole electrochemical field, morphology design and regulation has always been the primary way to improve electrode performances and the cyclic stability [33,34,35]. The common standing is that electrodes with high porosity or extensive grain boundaries are preferred, as they support rapid cation insertion/extraction [28,36]. However, optimization of the electrode morphology should be comprehensively balanced between the electrochemical characters and the cyclic stability. In addition, facing more sophisticated and harsh demands on electrochemical devices, more elaborate details urgently need to be taken into consideration during electrode designs, such as the granularity uniformity that usually goes along with porosity regulating of the film electrode.

Herein, the WO_3_ film electrode and the EC technology were taken as the research background where the WO_3_ films were prepared by magnetron sputtering. Instead of sputtering pressure or ratios of Ar/O_2,_ both of which could significantly regulate the film morphology [29,30], sputtering power was selected to minutely regulate the WO_3_ film morphology [31] and we depicted how such minute changes of the film morphology could play a pivotal role on the cyclic stability during cations’ insertion/extraction.

## 2. Experimental Section

### 2.1. Preparation of the Functional Films

Indium tin oxide (ITO)-coated glass substrates (1.1 mm thickness, ∼6 Ω sheet resistance) were purchased from Zhuhai Kaivo Optoelectronic Technology Co., Ltd. (Zhuhai, China), which were successively subjected to cleaning by acetone, alcohol, and deionized water before use. WO_3_ films were fabricated by direct current (DC) reactive magnetron sputtering using the JCP-500 sputtering system (Technol.cn) using a tungsten target with purity of 99.999%, 3 mm in thickness, and 3 inches in diameter. The target substrate separation was 110 mm. The background pressure was 6.0 × 10^−4^ Pa, and the working pressure was 1.4 Pa with an Ar flow rate of 16 sccm and an O_2_ flow rate of 6 sccm. The sputtering power was selected to tune the film morphology, which was set as 50 W, 80 W, 100 W, and 120 W. The substrate temperature was 25 °C. The deposition duration was 0.36~2.7 h to keep the film thickness constant. Meanwhile, the 1 mol L^−1^ Li ion electrolyte was prepared by dissolving LiClO_4_ in polycarbonate (PC) solution.

### 2.2. Characterization and Performance Tests

The film thickness was tested by a step profiler (Dektak XT Bruker), and all the prepared WO_3_ films were about 450 nm in thickness. Crystallization of the WO_3_ film was examined by X-ray diffraction (XRD) analysis using Cu Kα radiation (Philips X’Pert diffractometer). Surface and cross-sectional morphologies of the WO_3_ film were studied by scanning electron microscopy (SEM) using a Sigma 500 instrument (Zeiss). Tapping-mode atomic force microscopy (AFM) was performed using a Bruker Dimension Fast Scan to in situ monitor the morphology variation of the WO_3_ film during insertion and extraction of Li ions in the first EC cycle. Transmittance spectra were recorded using a UV−vis spectrophotometer (Shimadzu uv 3150, Japan). The cyclic stability was recorded by cyclic voltammetry (CV); the response time was measured by chronoamperometry measurement (CA), and diffusion coefficients of lithium ions in the WO_3_ films were analyzed from the (CV) characteristics at different scan rates. Adhesion between the WO_3_ film and the ITO glass substrate was tested by adhesive tape method, and the testing process was divided into three steps. First, transmittance of the WO_3_ film at 550 nm in the initial state (T_0_) was measured by UV/Vis spectrophotometer. Second, the ScotchTM adhesive tape at the force of 4.7 N/cm was leveled and tightly attached to the surface of the WO_3_ film. Then, the tape was torn up, and transmittance of the WO_3_ film at 550 nm was measured again (T_n_). The adhesion factor f was calculated according to the equation f = 1 − (T_n_ − T_0_)/(100 − T_0_). The closer the value of f was to 1, the greater the adhesion of the WO_3_ film was because at this time, almost no film was attached to the adhesive tape. On the contrary, the closer the value of f got to 0, the worse the adhesion of the WO_3_ film was. At this moment, the whole film was almost adhered to the adhesive tape. Finally, stress distribution in the WO_3_ film during Li ions’ insertion and extraction was analyzed by finite element simulation using the software package ABAQUS.

## 3. Results and Discussion

With the sputtering power ranging from 50 W to 120 W, the prepared WO_3_ films were all amorphous, which is preferred in application for the fast response and satisfactory coloration efficiency [31] (corresponding XRD spectra shown in Appendix A). Meanwhile, the surface morphology of the WO_3_ films showed minute but regular changes (Figure 1 and Appendix A). When the sputtering power was as low as 50 W (Figure 1a), relatively loose morphology was shown, containing both large, agglomerated clusters and fine particles, and the particle size centered at 28 nm with a wide range of 12.5~42.5 nm. For 80 W (Appendix A), the film became compact and was composed of nonuniform clusters, some parts with super-fine particles and others with big ones, and the particle size centered at 20 nm and 26 nm with a relatively narrow range of 14~38 nm. Up to 100 W (Figure 1b), the film seemed uniform and compact, and the particle size mostly centered between 14 nm and 17.5 nm with the narrowest range of 12~23 nm. When the sputtering power was further increased to 120 W, the film morphology turned back to the large, agglomerated morphology (Appendix A), along with a broad particle size distribution of 8~32 nm. According to the film growth mechanism, the higher the sputtering power, the bigger the kinetic energy and migrating capability of sputtered particles onto the substrate. Appropriate kinetic energy and migrating capability generally brings about uniform particle size and distribution; too much or too little would both result into particle aggregation and nonuniform granularity. Furthermore, cross-sectional SEM morphology also verified the relatively loose microstructure with nonuniform particle size distribution in the 50 W-deposited WO_3_ film and the dense packing morphology with uniform granularity in the 100 W-deposited film (Appendix A). Herein, the 50 W-deposited WO_3_ film and the 100 W-deposited WO_3_ film were selected, and their electrochemical behaviors and cyclic performances during cation insertion and extraction were recorded and systematically compared to unveil the intrinsic relationship among the porosity and granularity uniformity of the film electrode morphology, its electrochemical characteristics, and its cyclic stability.

As shown in Figure 2, response times of the two films at voltages of ±0.8 V for 20 s were measured by chronoamperometry, where the response time was taken as 90% of the current change [37]. The coloring and bleaching response times of the 50 W-deposited WO_3_ film were respectively 7.21 s and 2.51 s (Figure 2ai), while those of the 100 W-deposited film were 11.8 s and 5.26 s (Figure 2bi). This indicated that the 50 W-deposited film showed a faster response speed, originating from its relatively loose microstructure (Figure 1a) and enhanced transport dynamics of cations [32]. Moreover, the peak currents of the 50 W-deposited WO_3_ film during the coloring and bleaching processes were 10.3 mA and 26.8 mA, both higher than those of the 100 W-deposited film at 5.7 mA and 18.4 mA, verifying that lower charge transfer impedance was achieved, and Li ions’ insertion/extraction was facilitated when the film electrode was in relatively loose morphology.

After Li ions’ insertion and the WO_3_ film being fully colored, memory effects, i.e., the cation retention capability of the film electrodes, were recorded (Figure 2aii,bii). When the films were as-colored, transmittance modulation rates at the wavelength of 550 nm were 73% (from 78.3% to 5.3%) for the 50 W-deposited WO_3_ film and 76% (from 77.8% to 2.1%) for the 100 W-deposited film. After being exposed in the air for 24 h, the transmittance modulation rate decreased by 26% (from 73% to 47%) for the 50 W-deposited film and 21% (from 76% to 55%) for the 100 W-deposited film. Thus, a better cation retention capability would be achieved for the compact WO_3_ film, but optical performances of the two films were overall equivalent. In addition, in adhesive force tests with the ITO substrates, the two films also performed fairly, and the adhesion factors f were both 1, meaning excellent adhesive situations between the WO_3_ films and the ITO substrates were achieved from the magnetron sputtering method, which inherently provides good adhesion with substrates (Appendix A).

The electrochemical cyclic performances of the two WO_3_ films were analyzed based on their CV curves, shown in Figure 3. From Figure 3ai,bi and Appendix A (transmittance spectra evolution of the WO_3_ films during 5000 CV cycles), it could be seen that the transmittance modulation rate at 550 nm monotonically decreased from 73% to 37% after 5000 CV cycles for the 50 W-deposited film. Meanwhile, for the 100 W-deposited film, its transmittance modulation rate showed two sharp decreases during the first 2000 cycles, reducing from 76% to 69% during the first 1000 cycles and then to 61% during the second 1000 cycles, and then slightly declining to 58% in the following 3000 cycles, showing a better optical cyclic stability.

Charge density (mC·cm^−2^) can be obtained by integrating the CV curves (Appendix A) through Equation (1), where J, V, and s are the current density (mA·cm^−2^), voltage (V), and scanning rate (V·s^−1^), respectively.
(1)$$C=\int \frac{JdV}{s}$$

The calculated charge capacities of inserted (Q_in_), extracted (Q_ex_), and trapped (Q_ex/_Q_in)_ Li ions vs. the cycle number are shown in Figure 3aii,bii. In the beginning, the charge capabilities of the two films were both 2.88 mC·cm^−2^. For the 50 W-deposited film, it monotonically decreased to 1.04 mC·cm^−2^ after 5000 CV cycles, and the attenuation rate was calculated to be 64%; while for the 100 W-deposited WO_3_ film, the charge capacity decreased stage by stage, sharp declining from 2.73 mC·cm^−2^ to 1.72 mC·cm^−2^ during the second 1000 cycles, reaching a platform stage from the 2000th cycle to 3000th cycle, and then sharply declining to 1.18 mC·cm^−2^ in the last 2000 cycles. The attenuation rate was calculated as 59%, behaving slightly better than the 50 W-deposited film. It should be noted that during the 5000 CV cycles, variation trends of the optical modulation rate and the charge capacity of the two films were consistent.

However, as for the trapping and accumulation of Li ions in the films denoted by Q_ex/_Q_in_, for the 50 W-deposited WO_3_ film, Q_ex/_Q_in_ decreased from 69% to 66% at a very slow rate during the first 4000 cycles and then sharply declined to 55% during the last 1000 cycles, meaning that more Li ions were trapped at this time. Meanwhile, for the 100 W-deposited WO_3_ film, Q_ex/_Q_in_ decreased from 66% to 63% at a very slow rate during the first 3000 cycles, then sharply declined to 54% after 4000 cycles, and kept decreasing to 48% during the last 1000 cycles, which means much more Li ions were trapped in the 100 W-deposited film than the 50 W-deposited sample during the 5000 CV cycles because of its compact microstructure.

Furthermore, diffusion coefficients of Li ions (D) during the coloring and bleaching processes were analyzed according to the Randles–Servcik Equation (Equation (2)) from the CV characteristics at different scan rates v, illustrated in Figure 3aiii,biii (the original CV curves are shown in Appendix A, and analyzed evolutions of the redox peaks versus square root of the scanning rate are shown in Appendix A) [38].
(2)$$D^{1/2}=\frac{i_{Pc/Pa}}{2.69\times 10^{5}\times A\times C_{0}\times n^{3/2}\times v^{1/2}}$$
where *i_Pc_* and *i_Pa_* are the oxidation and reduction peak current intensities, A/cm^2^; *A* the effective area of electrochromic film, cm^2^; *v* is the scanning rate, V/s; *C*_0_ is the concentration of the cations in electrolyte, mol/cm^3^; and *n* is the number of involved electrons.

In the as-deposited state, the Li ions’ diffusion coefficient of the 50 W-deposited WO_3_ film were respectively 13.34 × 10^−10^ and 11.05 × 10^−10^ cm^2^/s in the coloring and bleaching process, which were slightly higher than those of the 100 W-deposited film of 12.89 × 10^−10^ and 10.7 × 10^−10^ cm^2^/s. This suggested that the faster response time of the 50 W-deposited film was probably because the interfacial transport of Li ions was facilitated rather than the bulk diffusion process, when the film was gifted a relatively loose morphology. During the 5000 CV cycles, diffusion coefficients both monotonously decreased for the two films because the more and more trapped Li ions would block the transport pathways in the film. However, the diffusion coefficients of the 50 W-deposited film slowly declined to 9.4 × 10^−10^ cm^2^/s during the coloring process and 5.61 × 10^−10^ cm^2^/s during the bleaching process; while for the 100 W-deposited film, they respectively decreased sharply to 4.4 × 10^−10^ cm^2^/s and 2.8 × 10^−10^ cm^2^/s after 5000 CV cycles, which suggested that morphology of the two films inevitably changed during such long-term Li ion insertion and extraction.

Thus, the surface morphology of the two film electrodes during the 5000 CV cycles were recorded and are shown in Figure 4. Overall, for both films, particles got finer and finer, which could also be responsible for the decreasing diffusion coefficients. However, the 50 W-deposited WO_3_ film finally cracked where the cluster profiles kept unchanged but the component particles had become finer during the first three cycles; after 1000 cycles, the component particles became even finer and the clusters seemed to be flattened with clearly seen borders filled with pores; after 3000 cycles, the film became nonuniform and was composed of relatively loose clusters mingled with some compact areas and several micro-cracks; after 5000 cycles, the film became very smooth and compact, and no particles and cluster profiles could be seen except huge cracks (Figure 4a). The morphology evolution could reasonably explain why Q_ex_/Q_in_ dropped sharply during the last 1000 cycles for the 50 W-deposited film (Figure 3aii). For the 100 W-deposited WO_3_ film (Figure 4b), similar variations happened to the morphology during the first three cycles; after 1000 cycles, cluster profiles blurred and the whole morphology was composed of a main compact area and some loose areas; while after 3000 cycles, the film returned to uniform and compact, and no more cluster profiles could be seen; after 5000 cycles, it still remained compact and uniform, and no mechanical damages were shown.

The morphology variation during the insertion and extraction of Li ions was recorded by in situ AFM (Figure 5), from which obvious different behaviors were observed for the two films. In the as-deposited state, particle cluster spikes could be seen on both surfaces of the two films, and their surface roughness values were similar at 8.09 nm for the 50 W-deposited film and 8.20 nm for the 100 W-deposited film. Once colored, the surface roughness values both decreased to 3.30 nm and 4.44 nm, respectively. However, for the particle cluster spikes, remarkable differences were observed in the two films. For the 50 W-deposited film (Figure 5aii), those cluster spikes became fat and were surrounded by gully regions. Once the Li ions started being extracted

(Figure 5aiii), although most of the cluster spikes returned to their as-deposited states with the surface roughness increasing to 6.20 nm, the film surface seemed full of bumps and gullies, even more rough than its as-deposited state. Contrarily, for the 100 W-deposited WO_3_ film, when Li ions were inserted (Figure 5bii), most cluster spikes disappeared, but some new small spikes appeared and dispersed well on the film surface. With Li ions being extracted (Figure 5biii), almost all spikes disappeared, and the surface roughness further decreased to 4.14 nm. These AFM results of the single process of Li ion insertion and extraction could be regarded a perfect epitome of the evolution of the film surface morphology during the 5000 CV cycles (Figure 4).

Ultimately, to deeply interpret the stress distribution in the WO_3_ film during cation insertion, finite element simulation was carried out using the software package ABAQUS, where the WO_3_ film was assumed to be perfectly bonded to a rigid substrate and the substrate was homogeneous and isotropic. Inhomogeneous bodies were employed to simulate the 50 W-deposited WO_3_ film with nonuniform particle size distribution (Figure 6ai), while the 100 W-deposited WO_3_ film was simulated as homogeneous and isotropic as the substrate (Figure 6bi). In this simulation, the WO_3_ film was supposed to be elastic, and the elastic modulus and the elastic properties of the inhomogeneous bodies were assumed to be much less than those of the uniform WO_3_ film. As can be seen from Figure 6bii, for the uniform 100 W-deposited film, stresses mainly centered in the middle of the film, but the two interface edges withstood a larger stress concentration. In this situation, under long-term cation insertion and extraction, the WO_3_ film more easily disconnected with the substrate and fell off as a whole. However, due to the nonuniform granularity of the 50 W-deposited film, stresses were much higher and mainly centered among boundaries of the inhomogeneous bodies. Additionally, the two interface edges also withstood a larger stress concentration than that of the uniform 100 W-deposited film. Therefore, under long-term cation insertion and extraction, cracks were more likely to generate in the WO_3_ film. The above simulation was verified by cross-sectional SEM images of the two WO_3_ films after 5000 CV cycles, as shown in Figure 6aiii,biii. In contrast with the 100 W-deposited film showing smooth and compact morphology, plenty of fragments and internal cracks were found in the 50 W-deposited WO_3_ film. It was interesting to note that during sample preparation of the SEM tests, it was hard to obtain smooth cross-sectional morphology for the 50 W-deposited film, and particle fragments always existed in the image both for the as-deposited state (Appendix A) and the 5000-cycled ones (Figure 6aiii), indicating weak bonding among the particles of the 50 W-deposited film. The strong stress distribution centered among cluster boundaries in combination with the weak bonding among particles were both responsible for its surface morphology variation, where the cluster spikes were surrounded by gullies after Li ion insertion and the cluster borders were filled with pores after 1000 cycles of Li ion insertion and extraction, until the final mechanical damage after 5000 CV cycles (Figure 4a and Figure 5a). However, for the 100 W-deposited film (Appendix A and Figure 6biii), smooth cross-sectional morphologies were easily obtained, suggesting the strong bonding among particles in the film, which perfectly explained why those cluster spikes were stretched by each other and disappeared during one-step Li ion insertion (Figure 5bii), and those scattered loose areas present in the 1000-cycled film would return to being uniform after 3000 CV cycles (Figure 4b).

## 4. Conclusions

In this work, the intrinsic relationship among the porosity and granularity uniformity of the film electrode morphology, its electrochemical characteristics, and the cyclic stability were demonstrated. Two kinds of WO_3_ films, one with a relatively loose morphology accompanied by nonuniform granularity and one with a compact morphology along with uniform particle size distribution were respectively obtained at the sputtering powers of 50 W and 100 W. As expected, owing to its relatively loose morphology, the 50 W-deposited film exhibited faster coloring and bleaching response times of 7.21 s and 2.51 s compared to the 100 W-deposited film, but had a worse capability of cation retention. The adhesive, electrochemical, and optical performances of the two film electrodes were equivalent in the beginning. However, during 5000 CV cycles, the cyclic stability of the 50 W-deposited WO_3_ film with cracked morphology and high transmittance modulation decay was far behind that of the 100 W-deposited film, though many more Li ions were trapped in the latter, known from its smaller Q_ex_/Q_in_ throughout. The strong stress distribution during Li ion insertion centering among boundaries of the nonuniform clusters and the weak bonding among particles in the relatively loose 50 W-deposited film together induced the final mechanical damage. These results demonstrated that although porosity gifts film electrodes fast kinetic characters, it also brings about weak bonding among active particles and weakens their mechanical cyclic stability during long-term cation insertion and extraction, and the granularity nonuniformity that generates strong local stresses would further aggravate such processes. Our study is an important attempt toward a more delicate design of the film electrode morphology, highlighting the trade-off between the kinetic behavior and the cyclic stability.

## Figures and Tables

**Figure 1 nanomaterials-13-00973-f001:**
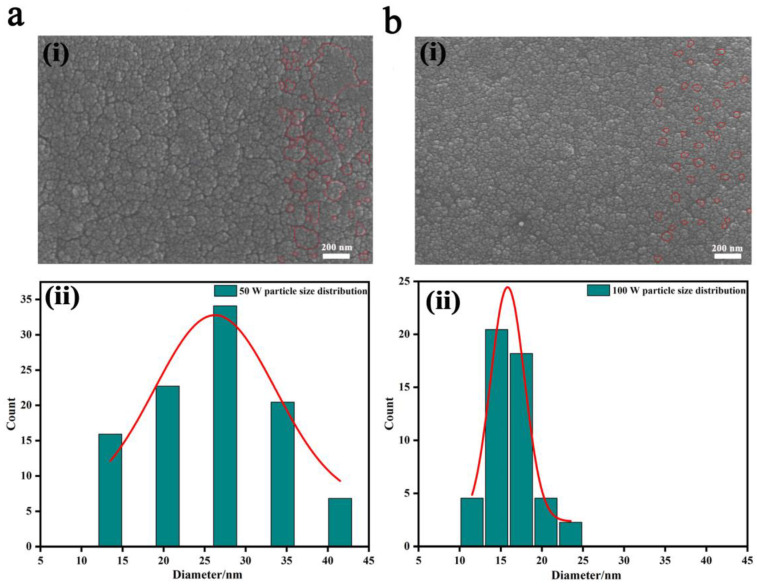
SEM images of the as-prepared WO_3_ films and corresponding particle size distribution analysis through Nano Measure software; (**a**) 50 W; (**b**) 100 W, (**ai**) and (**bi**) are SEM images, where red line circles in (**ai**) and (**bi**) picture the particle size distribution, (**aii**) and (**bii**) are the histogram of particle size distribution.

**Figure 2 nanomaterials-13-00973-f002:**
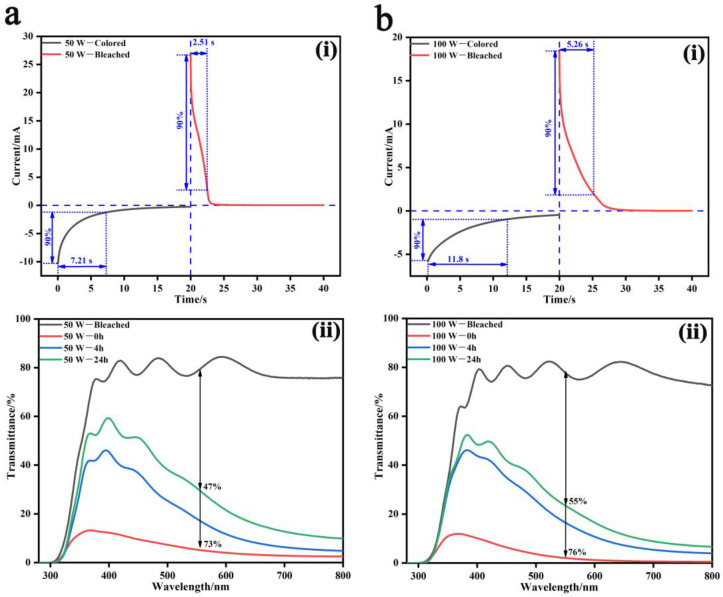
EC performances of the WO_3_ films prepared at the sputtering power of 50 W (**a**) and 100 W (**b**); response times (**ai**,**bi**), and memory effects (**aii**,**bii**) where the WO_3_ films were colored at 3 V for 5 min and then the transmittance spectra were recorded after different durations of 0 h, 4 h and 24 h.

**Figure 3 nanomaterials-13-00973-f003:**
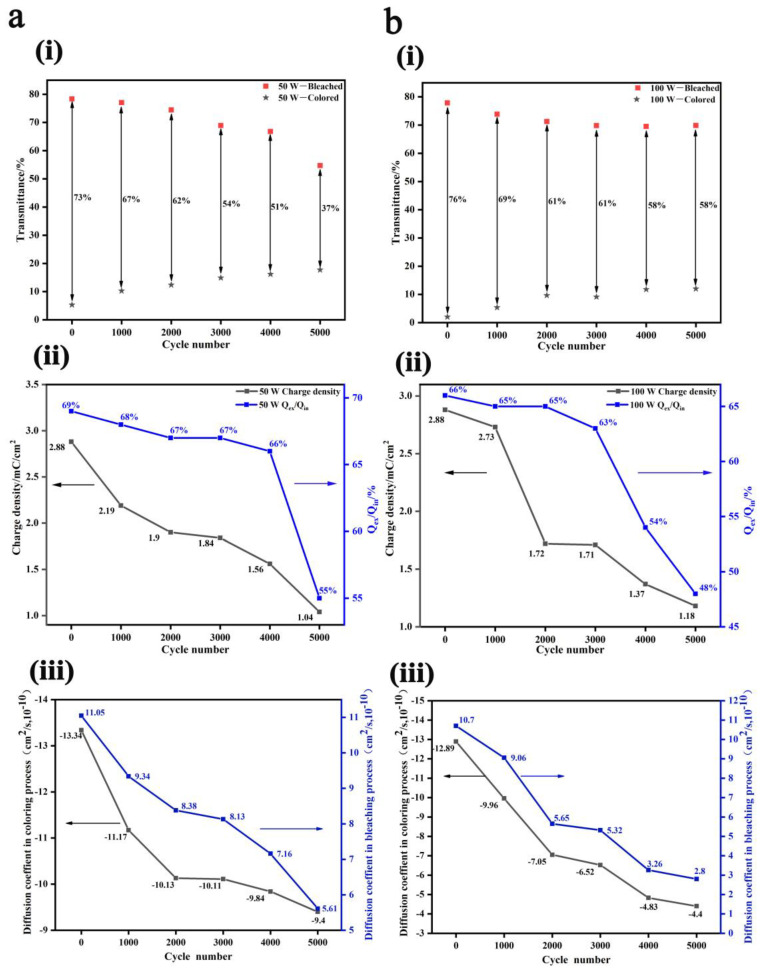
Electrochemical performances of the WO_3_ films during 5000 cyclic voltammetry (CV) cycles prepared at the sputtering power of 50 W (**a**) and 100 W (**b**), the transmittance modulation rate at wavelength of 550 nm (**ai**,**bi**), charge density attenuation and evolution of Li ion accumulation (**aii**,**bii**), diffusion coefficients of Li ions (**aiii**,**biii**).

**Figure 4 nanomaterials-13-00973-f004:**
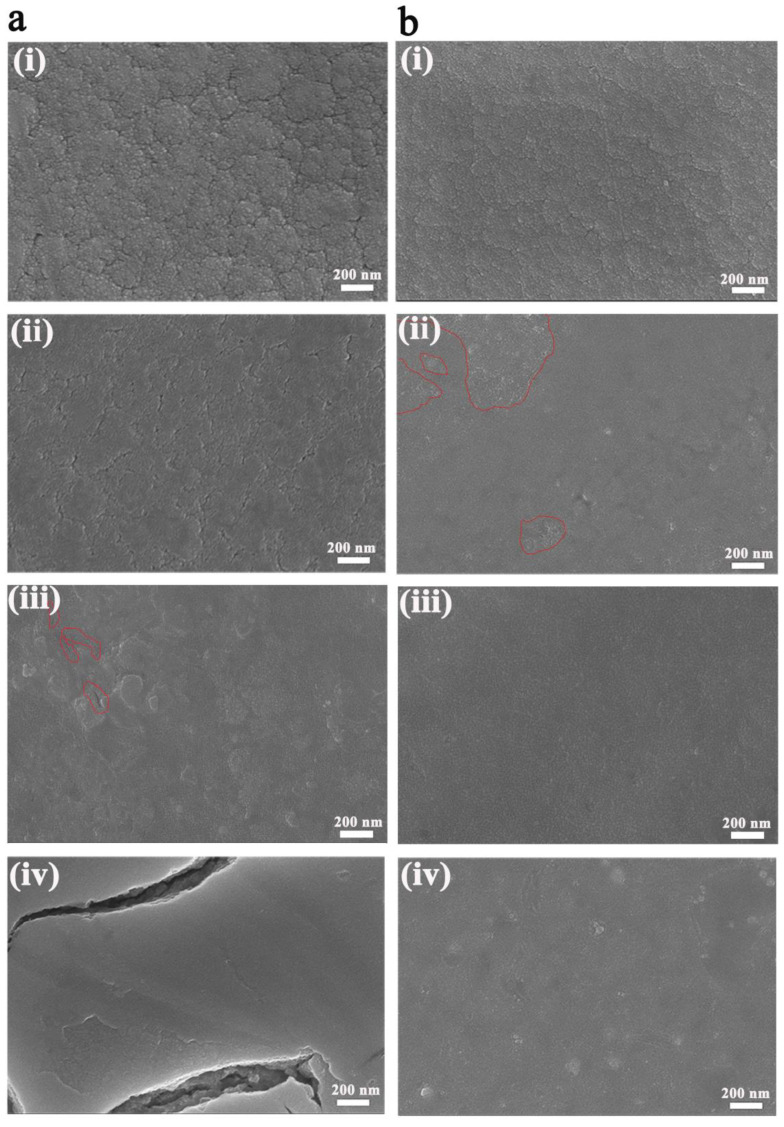
SEM morphology evolution of the WO_3_ films in bleached state during 5000 CV cycles prepared at the sputtering power of 50 W (**a**) and 100 W (**b**), after 3 cycles (**ai**,**bi**), after 1000 cycles (**aii**,**bii**), after 3000 cycles (**aiii**,**biii**), and after 5000 cycles (**aiv**,**biv**), where red line circles indicate the micro-cracks in the 50 W-deposited film and partly loose areas in the 100 W-deposited film.

**Figure 5 nanomaterials-13-00973-f005:**
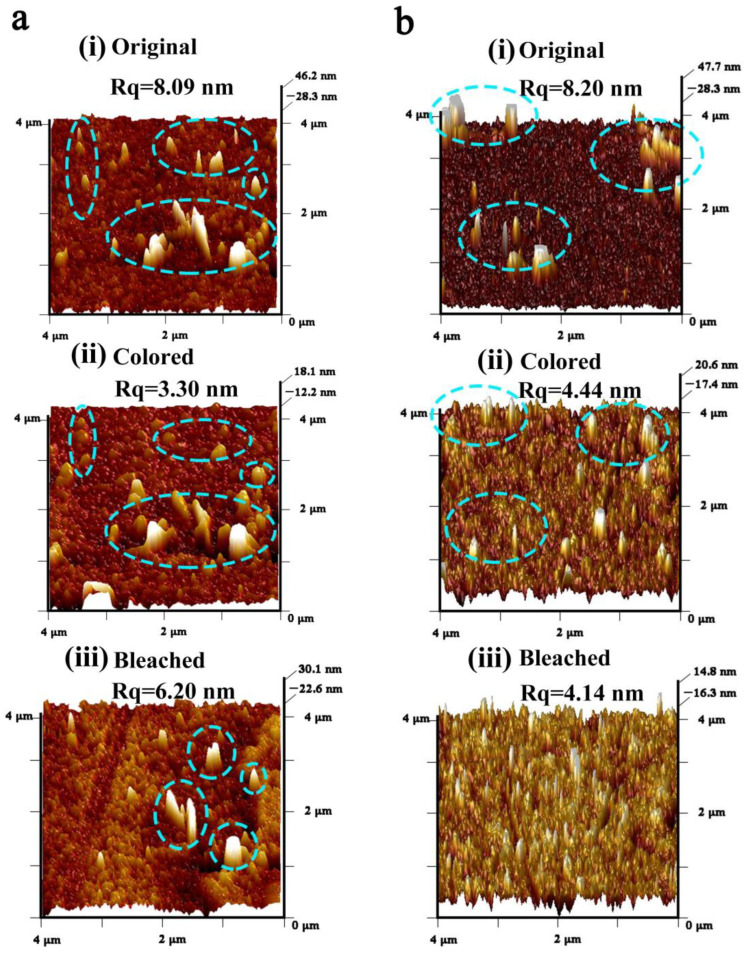
AFM morphology evolution of the WO_3_ films during one-step Li ion insertion and extraction prepared at the sputtering power of 50 W (**a**) and 100 W (**b**), the original state (**ai**,**bi**), the colored state (**aii**,**bii**), and the bleached state (**aiii**,**biii**), where blue line circles indicate the cluster spikes.

**Figure 6 nanomaterials-13-00973-f006:**
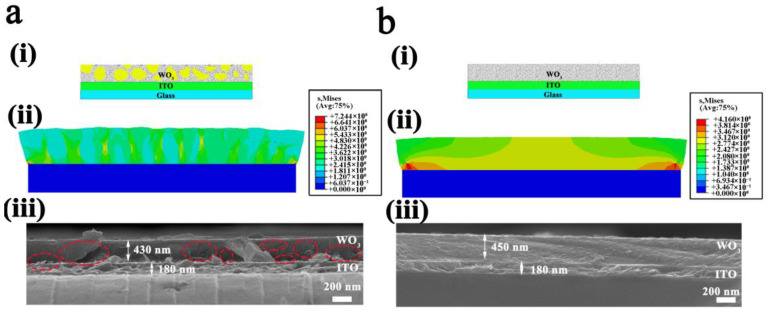
Failure analysis of the WO_3_ films with different granularity uniformity. Nonuniform state illustration of the 50 W-deposited WO_3_ film (**ai**) and uniform state illustration pf the 100 W-deposited WO_3_ film (**bi**), corresponding finite element simulation analysis (**aii**,**bii**), and cross-sectional SEM images of the WO_3_ films after 5000 CV cycles (**aiii**,**biii**), where red line circles indicate the cracks in the 50 W-deposited WO_3_ film.

## Data Availability

Data is unavailable due to the privacy of participants.

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
