# Peer review of "Pivotal Role of the Granularity Uniformity of the WO_3_ Film Electrode upon the Cyclic Stability during Cation Insertion/Extraction"

_nanomaterials, 2023, doi:10.3390/nano13060973_

Round 1
Reviewer 1 Report
The article discusses the features of obtaining thin-film electrodes based on tungsten oxide, obtained by sputtering under various conditions of preparation, as well as assessing the applicability of the obtained films as electrodes for batteries. The results of cyclic tests given in the article indicate a rather high prospect of using these materials as electrodes, while the authors carried out rather long cyclic tests, and also established the mechanisms that cause their destruction. In general, this article has a certain level of scientific novelty and practical significance, which makes it one of the candidates for publication in the selected journal. However, before making a final decision on this work, the authors should give a number of answers to the following questions.
1. The abstract presents information in sufficient detail on the prospects for using electrodes with different morphological features, but the reasons for choosing tungsten oxide as such materials were not given. The authors should give more details about the reasons for choosing such structures as objects for research.
2. The authors should clarify whether both samples had the same thickness, since the preparation conditions were significantly different, which could lead to inhomogeneities in the resulting thin films, as well as their thickness. When describing the objects of study, the authors should give more details.
3. The morphological features of the films presented in Figure 1 should be supplemented with more detailed images of the surface at high magnifications so that it is possible to separate the shape of the grains and the density of their fit to each other.
4. The authors should give the values of the changes in the transmission value during cyclic tests in a more visual form, since they are not informative in the presented form.
5. The main remark to the article is the lack of structural data on the phase composition of the obtained films under various conditions; in this case, the absence of these data does not allow one to unambiguously evaluate the observed changes and confirm the presented conclusions about the effect of stresses in the structure during cyclic tests. These stresses can also arise during lithiation, which are accompanied by interstitial processes and subsequent amorphization due to corrosion effects.
Reviewer 2 Report
Dear authors, thank you for submitting your manuscript entitled "Pivotal role of the granularity uniformity of the WO3 film electrode upon the cyclic stability during cations inserting/extracting" to MDPI Nanomaterials.
In your work, WO3 films, as the electrochromic film electrode, with a relatively-loose morphology accompanied by nonuniform granularity, and with a compact morphology along with uniform particle size distribution, were respectively obtained at the sputtering power of 50 W and 100 W. Electrochemical performances and cyclic stability of the two film electrodes were following-up traced and systematically compared.
Your manuscript appears to be suitable for publication following very minor revisions:
- Use vector graphics where possible -> directly export as vector graphic from Origin or as .EMF-file from powerpoint
- For figures with more or less everything images, increase the resolution
Good luck for your further research.
Reviewer 3 Report
This manuscript reports the studies on the electrochromic characters of the WO3 thin film synthesized by sputtering. The morphology was well characterized both for the as-prepared films and for the sample after 5000 CV cycles. The interesting results are shown not only in scientific but also in terms of practical use. Consequently, I would like to recommend the publication, if the following points are resolved.
1. Reference
Many papers have been published in WO3 films synthesized by sputtering. In the introduction, the previous researches should be referred and the difference from these reports should be provided.
Round 2
Reviewer 1 Report
The authors answered all the questions, the article can be accepted for publication.